# Evaluation of Radio Frequency-Assisted Enzymatic Extraction of Non-Anthocyanin Polyphenols from *Akebia trifoliata* Flowers and Their Biological Activities Using UPLC-PDA-TOF-ESI-MS and Chemometrics

**DOI:** 10.3390/foods11213410

**Published:** 2022-10-28

**Authors:** Xiaoyong Song, Yongli Jiang, Yu Zhong, Danfeng Wang, Yun Deng

**Affiliations:** 1College of Energy and Power Engineering, North China University of Water Resources and Electric Power, Zhengzhou 450011, China; 2Department of Food Science & Technology, Shanghai Jiao Tong University, 800 Dongchuan Road, Shanghai 200240, China; 3Shanghai Food Safety and Engineering Technology Research Center, Shanghai 200240, China

**Keywords:** *Akebia trifoliata* flower, radio frequency-assisted enzymatic extraction, non-anthocyanin polyphenols, anti-inflammatory, antibacterial activities

## Abstract

A new radio frequency heating-assisted enzymatic extraction (RF-E) method is applied for the determination of phenolic compounds in *Akebia trifoliata* flowers, compared with hot water, acidified ethanol (EtOH), and enzymatic-assisted (EA) extractions. Non-anthocyanin polyphenol profiles, antibacterial, angiotensin-converting enzyme (ACE) inhibitory, anti-inflammatory activities, and structures of extracts are evaluated. Results show no significant differences in the extraction of total flavonoid content (15.85–16.63 mg QEs/g) and ACE inhibitory activity (51.30–52.86%) between RF-E and EA extracts. RF-E extract shows the highest anti-inflammatory activities. FTIR and UV spectra reveal that acidified EtOH treatment has a significant effect on the structure of the extract due to its highest flavonoid content (20.33 mg QEs/g), thus it has the highest antibacterial activity against *Staphylococcus aureus* and *Escherichia coli.* Sixteen non-anthocyanin polyphenols are identified by UPLC-PDA-TOF-ESI-MS and RF pre-treatment did not cause significant compound degradation. The chemometric analysis shows that enzymatic hydrolysis significantly increased biological activities, and the presence of non-anthocyanin polyphenols correlates well with ACE inhibitory and anti-inflammatory activities. Accordingly, *A trifoliata* flowers have potential as reagents for the food and pharmaceutical industries due to their abundant polyphenols that could be extracted efficiently using RF-E.

## 1. Introduction

*Akebia trifoliata* is a Lardizabalaceae family member and a nutrient-rich semi-deciduous wild fruit tree distributed in China, Korea, Japan, and other Asian countries [1]. The roots, stems, and fruits of this species, as traditional Chinese medicines for over 2000 years, have effects on blood circulation, anti-inflammation, and diuresis [2]. However, *A. trifoliata* flowers have not been studied and are generally discarded as waste. Therefore, we examined whether these flowers could also be used as functional supplements.

Edible flowers contain a wide range of bioactive ingredients, including polyphenols that are extensively used in the food, pharmaceutical, and cosmetic industries due to their biological activities [3,4,5,6,7,8,9,10,11,12]. Both phenolic acids and flavonoids are important secondary metabolites derived from plants and are used in human therapeutics to treat degenerative disorders including cardiovascular disease, inflammation, and cancer [13,14]. The presence of these two groups of active metabolites is closely linked to flower color, and the purple color of *A. trifoliate* flowers suggests they contain abundant natural products and may have therapeutic potential [15,16]. 

In our previous study, we developed a novel technique, combining radio frequency and enzymatic hydrolysis (RF-E), to extract anthocyanin from *A. trifoliata* flowers. This method exhibited high crude yields and anthocyanin levels. Besides, we compared the antioxidant activities and enzyme inhibitory activities (i.e., α-amylase, tyrosinase, and protein tyrosine phosphatase 1B) of *A. trifoliata* flowers extracted with RF-E extraction and three conventional extraction methods, as well as identified their anthocyanin compositions. However, the anthocyanins accounted for only a small percentage of the *A. trifoliata* flower extracts. The extracts contained other more abundant non-anthocyanin polyphenols, including phenolic acids and other non-anthocyanin flavonoids that are linked to the biological activities of *A. trifoliata* flowers [8,17]. Therefore, it is essential to identify the non-anthocyanin polyphenols in the *A. trifoliata* flower. 

Besides, we further evaluated the effects of extraction techniques (i.e., hot water (HW) extraction, acidified EtOH (AE) extraction, enzymatic assisted (EA) extraction, and radio frequency heating-assisted enzymatic (RF-E) extraction) on the biological activities (i.e., antibacterial, angiotensin-converting enzyme (ACE) inhibitory, and anti-inflammatory activities) of the extracted, and evaluated the associations among the non-anthocyanin polyphenols with the biological activities. Our study describes the non-anthocyanin polyphenols present in *A. trifoliata* flowers for the first time. The results will provide a basis for further research of *A. trifoliata* flowers as a productive source of natural bioactive products for food and nutraceutical applications. 

## 2. Materials and Methods

### 2.1. Materials and Reagents

*A. trifoliata* flowers were purchased in April 2022 from Hunan Minghong Agriculture Development Co., Ltd. (Hunan Province, China). The flowers were freeze-dried and pulverized to a powder in a JP-500C electric grinder (Jiupindianqi, Yongkang, China). The powder was sieved through a 20-mesh screen and stored at −18 °C until use. All chemicals and reagents were of analytical grade and obtained from Shanghai Sinopharm Chemical Reagent Co., Ltd., Shanghai, China.

### 2.2. Extraction Procedures

Conventional extractions, including HW, AE, and EA extractions, were conducted as previously described in our previous study [1]. In brief, *A. trifoliata* flower powder was extracted with ultrapure water (1:20, *w*/*v*) at 100 °C for 30 min and then allowed to stand for 30 min. As for AE extraction, the flower powder was extracted with EtOH (1% HCl, *v*/*v*) at a ratio of 1:20 (*w*/*v*) for 60 min at 25 °C. As for EA extraction, the powder was extracted with 50% (*v*/*v*) aqueous EtOH (1:20, *w*/*v*) and 0.1% (*w*/*w*) of equal masses of pectinase and cellulase for 60 min (40 °C, pH 4.0). According to our previous study [1], the experimental conditions of RF-E extraction were as follows: 0.5% of cellulase and 0.5% of pectinase (*w*/*w A. trifoliata* powder), 50% of EtOH (*v*/*v*), pH at 4, a ratio of 1:20 (*w*/*v*), electrode gap of 5 cm. The mixture was heated by an RF heating device at 40 °C for 10 min. After RF pretreatment, the above enzyme mixture was added into the mixture and extracted at 40 °C for 50 min as per EA extraction.

After the extraction procedures, the mixtures were centrifuged for 10 min (6000× *g*, 4 °C). The solution was then evaporated at 40 °C and lyophilized except for the hot water extracts that were directly lyophilized without prior evaporation. The dried extracts were reconstituted in distilled water at different concentrations for analysis.

### 2.3. Determination of Total Phenolics and Flavonoids

Total phenolics content (TPC) and total flavonoid content (TFC) of extracts were determined as previously described by Jiang et al. [18]. Briefly, 0.2 mL of reconstituted extract solutions at 0.25 mg/mL was mixed with 1.0 mL of 10% Folin-Ciocalteu phenol reagent and allowed to stand for 3 min and 0.8 mL of 75 mg/mL Na_2_CO_3_ was added and the tube was shaken vigorously and incubated for 30 min in the dark at 37 °C. After cooling for 10 min at room temperature, the absorbance was measured at 765 nm using a Multiskan 1510 microplate reader (Thermo Fisher Scientific, Beijing, China). TPC was expressed as gallic acid equivalents according to the equation obtained from a standard curve generated using pure gallic acid. 

As for TPC, reconstituted extract solutions at 0.50 mg/mL were mixed with an equal volume of 2% AlCl_3_
*w*/*v* in methanol and allowed to stand for 20 min. The absorbance of the solutions was then measured at 415 nm against a blank sample with AlCl_3_ using a microplate reader. TPC was expressed as gallic acid equivalents according to the equation obtained from a standard curve generated using pure gallic acid. TFC was expressed as equivalents of quercetin by comparison with a standard curve generated using pure quercetin.

### 2.4. Identification of Non-Anthocyanin Polyphenols by UPLC-MS/MS

Phenolic compounds were identified by the Shanghai Jiao Tong University Analysis using an Acquity I-class UPLC system (Knauer Smartline, Berlin, Germany) equipped with UPLC BEH Amide column (1.7 μm, 100 × 2.1 mm) (Waters, Burlington, MA, USA). The parameters were as follows: column temperature at 45 °C, injection volume of 1 μL; flow rate at 0.4 mL/min. The mobile phases consisted of (A) 0.1% formic acid in ultrapure water (*v*/*v*) and (B) 0.1% formic acid in acetonitrile (*v*/*v*). The elution gradient was conducted as follows: 0 min, 95% A; 3 min, 80% A; 10 min, 0% A; 12 min, 0% A; 15 min, 5% A; 19 min, 5% A. Mass-Lynx TM V 4.1 software (Waters, Milford, MA, USA) was used to collect the mass spectra in negative ion mode. The MS parameters were as follows: capillary voltage, 2.0 kV; sampling cone voltage, 40 V; extraction cone voltage, 2.0 V; desolvation temperature, 450 °C; cone gas flow, 50 L/h; collision energy, 20.0–45.0 eV; run time, 20.00 min; scan time, 0.50 s.

### 2.5. Structure Analysis

UV-visible spectra of reconstituted extract solutions at 0.1 mg/mL were scanned from 200 to 500 nm and wavelength shifts at maximum absorbance were determined as previously described [19]. FTIR spectra were obtained using lyophilized powders (1–2 mg) that were diluted by grinding with KBr and pressing into pellets. FTIR spectra were recorded using a Nicolet 6700 spectrometer (Thermo Nicolet, Madison, WI, USA) using a total of 16 scans from 400 to 4000 cm^−1^ against the background taken at 4 cm^−1^ resolution.

### 2.6. Biological Activities

#### 2.6.1. Angiotensin-I Converting Enzyme (ACE) Inhibitory Activity

Reconstituted extract solutions at 0.1 mg/mL were used to evaluate the ACE inhibitory activity using a CK-E11125H commercial kit (Dojindo Chemical Technology Co., Ltd., Shanghai, China) as previously described [20]. Absorbance was measured using a microplate reader at 450 nm (see above). ACE inhibitory activity was calculated as follows: $$ACE\;inhibitory\;\left(\%\right)=\left(\frac{A_{blank1}-A_{Sample}}{A_{blank1}-A_{blank2}}\right)\times 100$$
where *A_blank_*_1_ was the absorbance of control wells, *A_sample_* was the absorbance of sample-treated wells, and *A_blank_*_2_ was the absorbance of blank wells.

#### 2.6.2. In Vitro Anti-Inflammatory Activities

Reconstituted extract solutions at 0.1 mg/mL were also examined for their in vitro anti-inflammatory activity, including human granulocyte-macrophage colony-stimulating factor (GM-CSF) and tumor necrosis factor β (TNF-β) using commercial enzyme-linked immunosorbent assays (Dojindo Chemical Technology Co., Ltd., Shanghai, China) as described [21]. Absorbance was measured using a microplate reader at 450 nm and readings were interpolated from standard curves as suggested by the manufacturer.

#### 2.6.3. Antibacterial Activities

Antibacterial activities of extracts were measured against *Escherichia coli* (CICC 10003) and *Staphylococcus aureus* (CICC 10384). The loops of *E. coli* and *S. aureus* were inoculated into 50 mL Nutrient Broth to incubate by gentle shaking at 120 rpm for 12 h at 37 °C until the end of the exponential phase of growth. After incubation, the bacterial suspensions were standardized by adjusting based on 0.5 McFarland, which corresponds to 0.08–0.10 absorbance (~10^8^ CFU/mL) using a microplate reader at 600 nm. 

The 4 mL of diluted suspensions were mixed with 2 mL of reconstituted extract solutions at different concentrations (0.10, 0.25, 0.50, 1.00, and 2.00 mg/mL). A tube of 4 mL of diluted suspensions with 2 mL of 0.9% saline antimicrobial agent has served as growth control. The absorbance at 600 nm was measured after incubation for 24 h at 37 °C. To evaluate the effect of incubated time on the bacterial growth trend, the absorbance of suspension aliquots (1.00 mg/mL of reconstituted extract solutions) at 600 nm were measured at 0, 1, 2, 3, 7, 11, 24, and 30 h. 

In addition, the agar diffusion method was also used for determining the antibacterial activities of extracts as previously described [22].

### 2.7. Statistical Analysis

All experiments were conducted at least three replicates and the data were expressed as the mean ± standard deviation. Data were analyzed by ANOVA and significant differences at *p* < 0.05 were determined using Duncan’s multiple range tests with SPSS 21.0 statistical data analytical software (IBM, New York, NY, USA). 

## 3. Results and Discussion

### 3.1. TPC and TFC

We initially evaluated the phenolic and flavonoid contents of *A. trifoliata* flower extracts using techniques appropriate for further applications in food processing. As shown in Figure 1A, the extracts generated were different for TPC and varied from 121.24 to 165.26 mg GAEs/g extract. These TPC levels exceeded those for 51 flowers previously examined, indicating *A. trifoliata* flowers are a rich source of bioactive compounds [16]. The maximum TPC was observed for EA extraction and was slightly less for RF-E (165.26 and 144.67 mg GAEs/g extract, respectively). This difference was most likely related to the degradation of thermolabile bioactive compounds during RF treatment [23,24]. In contrast, the AE extract contained the highest level of TFC (20.33 mg QEs/g extract), while the lowest was observed for HW extraction (10.47 mg QEs/g extract). We found no significant (*p* > 0.05) TFC differences between the RF-E and EA extracts. 

The level of bioactive compounds in plant materials is closely linked to the method of extraction. Our previous study concluded that HW extraction could improve crude yields, but this also significantly reduced flavonoid levels due to prolonged heat treatment [1]. In addition, impurities that were insoluble at ambient temperature may be dissolved in ethanol at elevated temperatures and inhibit or interfere with bioactivity [13]. It was found that AE extraction had negative effects on the crude yields from *A. trifoliata* flowers. In contrast, the present EA extraction benefited from lower temperatures, and TPC was increased. In addition, ethanol is more efficient than water in the extraction of flavonoids due to a decrease in polarity [17]. Moreover, acidification and lower extraction temperatures had a positive effect on flavonoid stability. 

### 3.2. UPLC-MS Analysis

The anthocyanin compositions of *A. trifoliata* flowers were also identified [1]. Considering the possible other bioactive compounds present, we identified the non-anthocyanin polyphenols extracted with different extraction methods. As shown in Figure 2, the UPLC chromatograms of the RF-E extract and EA extract were very similar. The AE extract was similar to the HW extract except for peak height differences. We identified 16 non-anthocyanin polyphenols that included chlorogenic acid, caffeic acid, 3-feruloylquinic acid, isochlorogenic acid, quercetin, and kaempferol. The molecular structures of these six compounds were also confirmed by their mass measurements and fragmentation patterns. The primary components by mass were the hydroxycinnamic acids and flavonols. Chlorogenic acid and derivatives were the major phenolic acids, and the flavonols consisted primarily of quercetin, kaempferol, and their derivatives, which were similar to the composition of blackthorn flowers [25]. This is the first time non-anthocyanin polyphenols have been isolated and identified from *A. trifoliata* flowers. Their presence explains and supports the use of *A. trifoliata* flowers in traditional medicine.

Generally, the four extraction procedures yielded significant differences in the content of bioactive compounds. Most compounds were more abundant in the RF-E and EA extracts, and isochlorogenic acid levels were 4-fold greater than those in the AE extract and HW extract. In addition, the AE extracts contained the highest levels of quercetin and kaempferol, while the flavonols were more abundant in RF-E extract and EA extract. These results indicated that enzymatic hydrolysis released greater levels of phenolic acids while acidified EtOH preferentially extracted flavonoids as shown by TFC and TPC values (Figure 1A). Similarly, the hydroxycinnamic acid and total non-anthocyanin polyphenol content of RF-E extract and EA extract were higher than those of AE and HW extracts (Table 1). Overall, these results indicated that RF coupled with enzymatic hydrolysis was a reliable method for non-anthocyanin polyphenol extraction that primarily consisted of hydroxycinnamic acids and flavonols. These non-anthocyanin polyphenols are more stable at high temperatures compared with the destructive effects of heat on *A. trifoliata* flower polyphenols (i.e., HW extraction). These results indicated that non-anthocyanin polyphenols were more stable at higher temperatures [15].

### 3.3. UV and FTIR 

The UV spectra of extracts treated using different extraction methods all displayed typical broad peaks at 250–300 nm related to phenolic compounds. As shown in Figure 3A, the UV spectra were not significantly correlated with TPC (Figure 1A), which might be associated with other compounds such as tyrosine (278 nm) and tryptophan (279 nm) absorbing at that wavelength [30]. The increase in absorbance intensity for the RF-E extracts over the EA and AE extracts indicated that RF treatment induced molecular unfolding by damaging interactions of adjacent protein molecules, thus exposing more Tyr and Phe residues [23]. The AE extracts displayed a significant absorbance peak at 300–350 nm that correlated with the TFC (Figure 3A). 

The structure or composition of the extracts can be identified based on the differences in absorption frequencies of different functional groups or chemical bonds using FTIR as previously described [15,26]. As shown in Figure 3B, all our extracts showed a large absorption peak at 3413.39–3428.81 cm^–1^ that was attributed to –OH group stretching [26]. The weak peak at 2850.27–2875.34 cm^−1^ corresponded to the C–H stretching of the carbohydrate methylene and was enhanced after AE treatment. The peak at 1529.27 cm^−1^ was primarily due to amide II (40% CN stretch and 60% NH bend) that moved to amide I between 1631.48 and 1733.69 cm^−1^ (C=O stretching) after AE treatment. This indicated that AE had a significant effect on proteins in the extracts, most likely due to the strongly acidic conditions used for extraction. The peak at 1384.46–1402.00 cm^−1^ represented deformation and stretching vibrations of C–C bonds in phenolics, while 1076.08–1116.58 cm^−1^ and 609.40–611.32 cm^−1^ were assigned to C=C groups and aromatic rings, respectively [19]. Overall, we found significant differences between the AE extract and the other extracts, but no significant difference between RF-E and EA extracts (Figure 3B).

### 3.4. ACE Inhibitory Activity

ACE inhibition is an indicator of the presence of an overall anti-hypertensive effect due to effects on water and salt metabolism and the inactivation of the hypotensive peptide bradykinin [20]. All extracts displayed ACE inhibitory activities ranging from 39.58 to 52.86%, which were the greatest for the enzyme treatment groups (Figure 1B). Overall, these values were greater than for ACE inhibition derived from methanol extracts of bamboo shoots [7]. It can be seen from Figure 1B that the enzyme treatment groups (RF-E and EA extracts) showed significantly (*p* ˂ 0.05) higher ACE inhibitory activity. A strong relationship between ACE inhibitory activity and TPC has been reported in the literature [14]. Moreover, enzymatic hydrolysis might increase protein extraction, thus resulting in enhanced release of ACE inhibitory peptides [26]. However, we found no significant (*p* > 0.05) difference between AE and HW extracts, in contrast to a previous study that found that methanol extraction had significantly (*p* ˂ 0.05) higher ACE inhibitory activity than water extraction [7]. This phenomenon might be explained by the fact that the acidic condition of AE extraction was not suitable for the ACE inhibitory peptide extraction [21].

### 3.5. In Vitro Anti-Inflammatory Activities

Anti-inflammatory activities play an important role in both food systems and the human body in reducing inflammatory responses. GM-CSF and TNF-β are commonly used to evaluate in vitro anti-inflammatory activities. Higher values indicate higher anti-inflammatory activities and the presence of greater numbers of anti-inflammatory molecules [21]. Similar to ACE inhibitory activity results, both GM-CSF and TNF-β levels in the RF-E extract and EA extract were greater than those in the AE extract and HW extract. In addition, the RF-E extract exhibited the highest GM-CSF and TNF-β values (0.61 and 0.59 mg/kg, respectively). Enzymatic hydrolysis significantly induced cytokine production, and RF pretreatment further improved the positive effects. Non-anthocyanin polyphenols such as quercetin or resveratrol, as well as plant polyphenol extracts from green tea and grape seed, have been reported to increase anti-inflammatory cytokines in colitis models [31]. We did find abundant levels of quercetin and derivatives in *A. trifoliata* flowers, especially in the RF-E and EA extracts (Figure 1B). These results indicated that extracts from *A. trifoliata* flowers could be immunomodulatory and protect the colon from inflammatory disorders due to their high non-anthocyanin polyphenol content. 

### 3.6. Antibacterial Activities

The antibacterial activities of our extracts were examined by their ability to inhibit bacterial proliferation over a 24 h period. All extracts showed remarkable antibacterial activities against *S. aureus* and *E. coli*, and the antibacterial activities increased with increasing concentrations. (Figure 4A,B). There were no significant (*p* > 0.05) differences between HW and RF-E extracts for inhibiting *E. coli* growth at low concentrations (˂1.00 mg/mL), while RF-E extracts showed higher inhibitory activity at 2.00 mg/mL. In addition, AE extract exhibited the highest antibacterial activity against *S. aureus* and *E. coli*, which might be associated with the highest TFC content (Figure 1A). Casagrande et al. [32] also reported that ethanolic extract has well-known isolated bioactive components such as borneol, camphor, carvacrol methyl ether, and methyl palmitate, thus increasing the antibacterial activity.

To evaluate the effect of incubation time on the bacterial growth trend, the absorbance of suspension aliquots (1.00 mg/mL of reconstituted extract solutions) at 600 nm were measured at different incubation times. The results are presented in Figure 4C,D. As could be seen, the absorbances of all groups were increased, indicating that the bacteria were continuously growing with time. In the beginning, the absorbances of the extracts groups were higher than those of the control group due to the bioactive compounds in extracts. The absorbances of the control group increased faster than those of the extracts groups, indicating that the addition of extracts could significantly inhibit the growth rate of the bacteria (Figure 4C,D). After 24 h, the absorbance of the control group was over 0.80, and the inhibition rates against *E. coli* ranged from 20.10% (HW) to 42.09% (AE). Besides, extracts had a stronger inhibitory effect against *S. aureus* than against *E. coli,* with higher inhibition rates (39.64–63.70%) at the same time. This might be associated with the fact that gram-negative bacteria possess an outer layer of the lipopolysaccharide cell membrane that makes them more protected [29]. 

The diameters of the growth inhibition zones due to the effects of different extracts from *A. trifoliata* flowers on *S. aureus* and *E. coli* were shown in Figure 4E. It was observed that all the extracts had an antibacterial effect on *S. aureus* and *E. coli*. Non-significant (*p* > 0.05) differences in the inhibition zone of *E. coli* were observed among the RF-E, EA, and AE extracts, which were higher than that of the HW extract. Similarly, the inhibition zone of *S. aureus* of the HW extract was the lowest (9.75 mm) but was significantly higher than that of *E. coli* (6.83 mm), indicating a stronger inhibitory effect against *S. aureus.* Furthermore, the RF-E extract showed the highest inhibition zone diameter of *S. aureus* (12.00 mm), significantly higher than that of other extracts. One important factor affecting the antibacterial activity is the difference in the bioactive compositions of extracts, which are influenced by the growth stage, method of drying, and extraction technique. This study was the first to experimentally assess the antibacterial effect of *A. trifoliata* flower extracts. The results in this study indicated that *A. trifoliata* flowers are a potential source of antibacterial agents, and extracts by acidified EtOH treatment showed greater antibacterial activity. 

### 3.7. Chemometrics Analysis

#### 3.7.1. Principal Component Analysis (PCA) and Heatmap Analysis

There were many data indicators to evaluate the properties of *A. trifoliata* flower extracts, so we applied principal component analysis (PCA) to reduce the dimensions of the data. As shown in Figure 1A, the relation among these variables could be reduced to two components that explain 89.9% of the total variation (PC1 accounts for 69.4% and PC2 for 20.5%) (Figure 5A). As expected, extracts from different extraction methods present distinct separations, and the three replicates of each group have similar PC scores, indicating that the four extracts present distinct variations and show small separations among replicates. These results showed that the RF-E extract and EA extract were closely linked and comprised one classification (Figure 5A). A heatmap analysis was applied to the effect of extraction techniques on the physicochemical properties of *A. trifoliata* flowers (Figure 5B). Each column represents a flower extract, and each row represents an experiment index. Cluster analysis based on the Pearson correlation coefficient suggests that the included samples can be divided into two groups. The results agreed with the PCA analysis, and the AE extract was significantly different from other extracts.

#### 3.7.2. Orthogonal Partial Least Squares-Discriminant Analysis (OPLS-DA)

To better study the differences between these *A. trifoliata* flower extracts, we used the OPLS-DA model to study the differences between Class 1 (RF-E, EA, and HW extracts) and Class 2 (AE extract) (Figure 5C). A sample score plot indicated a clear separation between the two classifications. The R^2^Y and Q^2^ of the model were 99.6% and 99.1%, respectively, and we then performed 200 substitutions to further verify the predictability of the OPLS-DA model. The cross-validation with 200 permutation tests indicated that this OPLSDA model was reliable, with intercepts of R^2^ and Q^2^ being 0.24 and −1.1, respectively. Additionally, all points on the left were lower than those on the right (Figure 5D), indicating a significant degree of predictability [33]. 

An S-plot was drawn to show the effect of sensitive indicators for the *A. trifoliata* flower extracts (Figure 5E). The indicators in the upper right and lower left corners of the S-plot were considered discriminant markers for selecting potential indicators [34]. The corresponding loading plot showed that TFC, TPC, ABTS, GM-CSF, TNF-β, ACE inhibitory, and CUPRAC were considered sensitive markers, which were the primary indicators that affect the *A. trifoliata* flower extracts’ biological activities. 

### 3.8. Correlation Analysis

A correlation-based approach was adopted in this study to evaluate the associations among the non-anthocyanin polyphenols with biological activities. Compounds **4** and **8** had significant (*p* < 0.05) anti-inflammatory activities due to increased levels of GM-CSF and TNF-β. Some hydroxycinnamic acids (compounds **2**, **3**, **5**, **6**) and flavonols (compound **15**) were significantly positively correlated with GM-CSF. Similar to the result of anti-inflammatory, five hydroxycinnamic acids (compounds **3**–**6**) were also significantly (*p* < 0.05) positively correlated with ACE inhibitory. In addition, the remaining **7** compounds, including chlorogenic acid (compound **1**) and isochlorogenic acid (compound **12**), as well as quercetin and kaempferol derivatives (compounds **7**, **10**, **11**, **13**, and **14**), were significantly correlated with ACE inhibitory activity (Table 2). Total hydroxycinnamic acids and non-anthocyanin polyphenols had significant (*p* < 0.10) impacts on ACE inhibitory and anti-inflammatory activities, but the total phenolics and flavonoids were not significantly (*p* > 0.05) correlated with these two biological activities. For antioxidant activities, compounds **9**, **12**, **14,** and flavonols were significantly (*p* < 0.10) correlated with DPPH, whereas there were other non-anthocyanin polyphenols correlated with ABTS (compound **2**), CUPRAC (compounds **1**, **7**, and **11**), and RP (compounds **8**, **9**). Flavonols content had significant (*p* < 0.10) impacts on DPPH and RP. Additionally, TFC was significantly (*p* < 0.10) correlated with CUPRAC. Different from the above biological activities, there was only one non-anthocyanin polyphenol (compounds **9**) correlated with enzyme inhibitory activities (Tyrosinase and PTP-1B). 

Moreover, none of the 16 non-anthocyanin polyphenols were correlated with antibacterial activities for *S. aureus* and *E. coli*. However, antibacterial activities were significantly (*p* < 0.10) correlated with the total flavonoid content and flavonol content but not significantly correlated with total phenolics. A previous study indicated the absence of any significant bivariate correlations between phenolics and growth inhibition of *E. coli* [35]. Some anthocyanins extracted from *Khaya senegalensis* were also found to possess antibacterial activities [29]. These results indicated that antibacterial activities were dependent on the collective effect of the bioactive compounds and not dependent on a particular compound.

## 4. Conclusions

In this study, HW extraction generated the lowest yields of bioactive compounds and biological activities due to prolonged heating. The AE extracts were associated with the highest TFC and antibacterial activities. Enzymatic hydrolysis significantly (*p* < 0.05) improved the TPC, ACE inhibitory, and anti-inflammatory activities. RF pretreatment resulted in a slight decrease in phenolics but still exhibited the highest anti-inflammatory activities. To the best of our knowledge, our study for the first time provides a description of the non-anthocyanin polyphenols present in *A. trifoliata* flowers. We identified 16 non-anthocyanin polyphenols, including chlorogenic acid, quercetin, kaempferol, and their derivatives. The PCA and OPLS-DA analyses showed that RF-E and EA extract comprised one classification, which had higher levels of phenolic acids, while acidified EtOH treatment was beneficial in extracting flavonoids. The non-anthocyanin polyphenols possessed significant ACE inhibitory and anti-inflammatory activities, while TFC and flavonols were significantly (*p* < 0.05) correlated with the antibacterial activities of *A trifoliata* flower extracts. 

## Figures and Tables

**Figure 1 foods-11-03410-f001:**
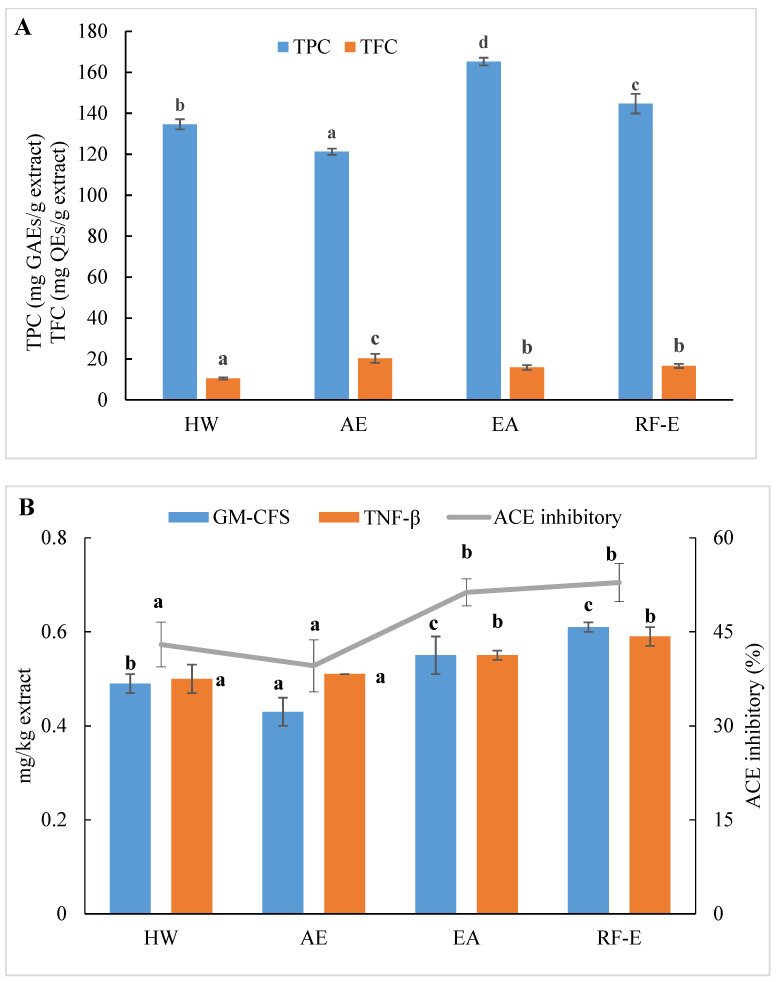
Effects of RF-E and conventional extractions of (**A**) phenolic compound levels and (**B**) biological activities of *A. trifoliata* flower extracts. Mean ± standard deviation (*n* = 3). Different lowercase letters in the same color column show significant differences at the *p* < 0.05 determined by Duncan’ s multiple range tests. HW, hot water extract; AE, acidified ethanol extract; EA, enzymatic assisted extract; RF-E, radio frequency dielectric heating-assisted enzymatic extract; TPC, total phenolic; TFC, total flavonoids; GAE, gallic acid equivalents; QE, quercetin equivalents; ACE, Angiotensin-I converting enzyme; GM-CSF, granulocyte-macrophage colony-stimulating factor; TNF-β, tumor necrosis factor β.

**Figure 2 foods-11-03410-f002:**
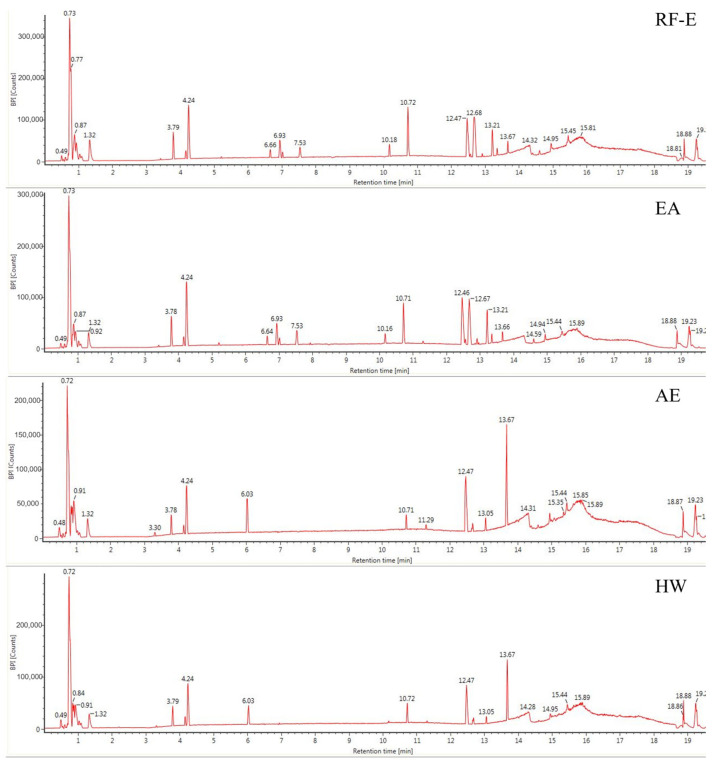
Base peak chromatograms of *A. trifoliata* flower extracts. See Figure 1 for abbreviations.

**Figure 3 foods-11-03410-f003:**
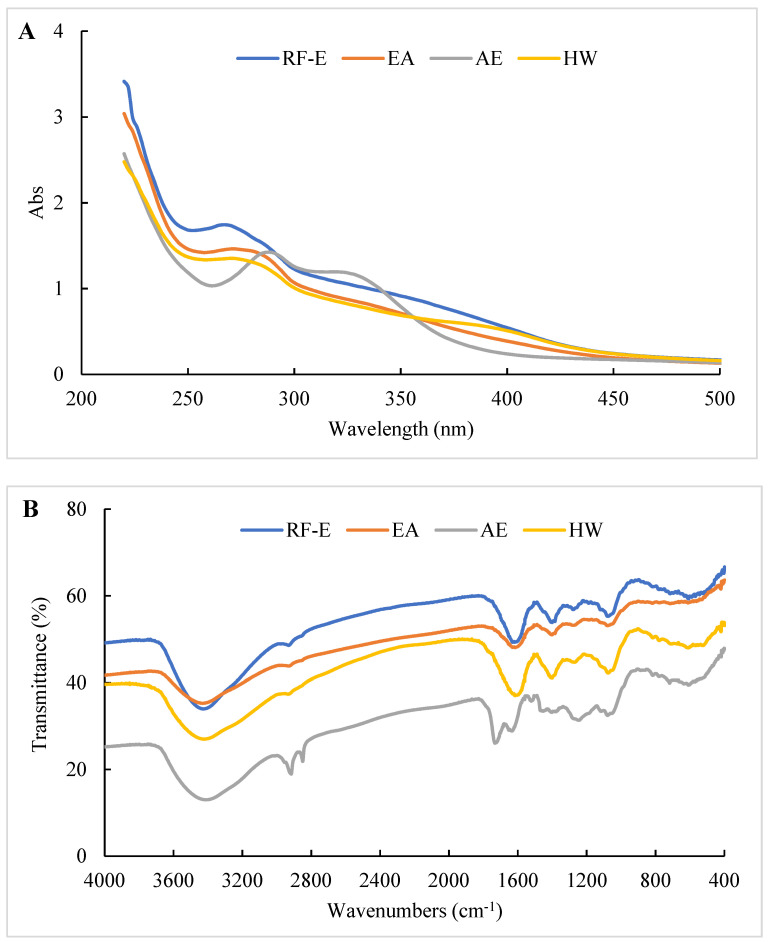
UV (**A**) and FTIR (**B**) spectrophotometric analysis of *A. trifoliata* flower extracts. See Figure 1 for abbreviations.

**Figure 4 foods-11-03410-f004:**
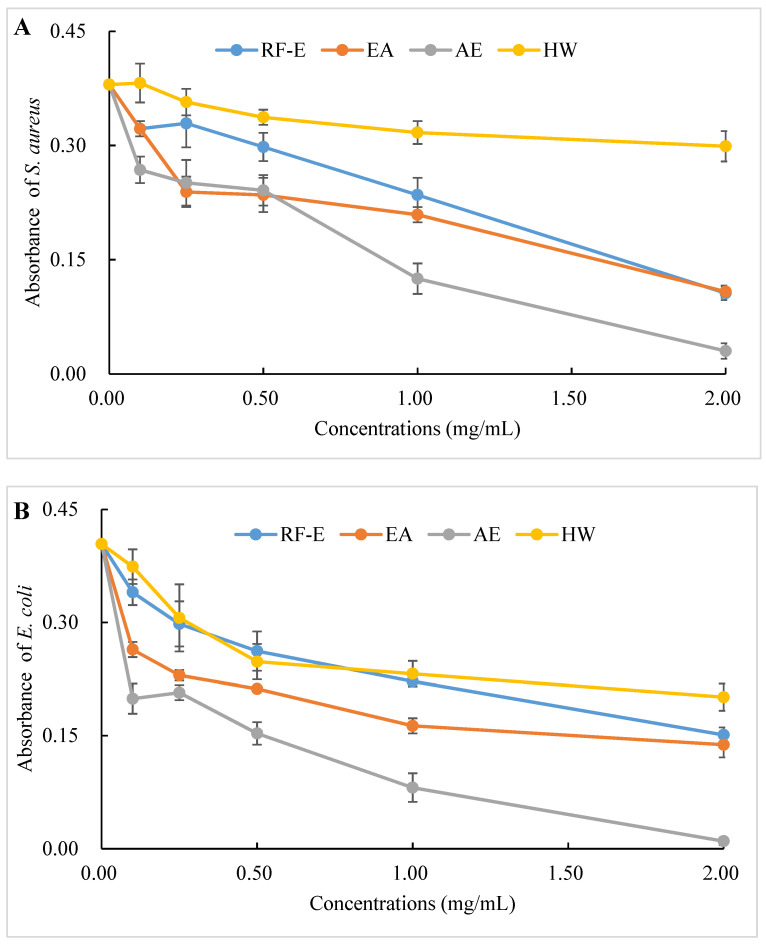
Antibacterial activities of *A. trifoliata* flower extracts. Different lowercase letters show significant differences at the *p* < 0.05 determined by Duncan’ s multiple range tests. See Figure 1 for abbreviations. *S. aureus*, *Staphylococcus aureus*; *E. coli*, *Escherichia coli*.

**Figure 5 foods-11-03410-f005:**
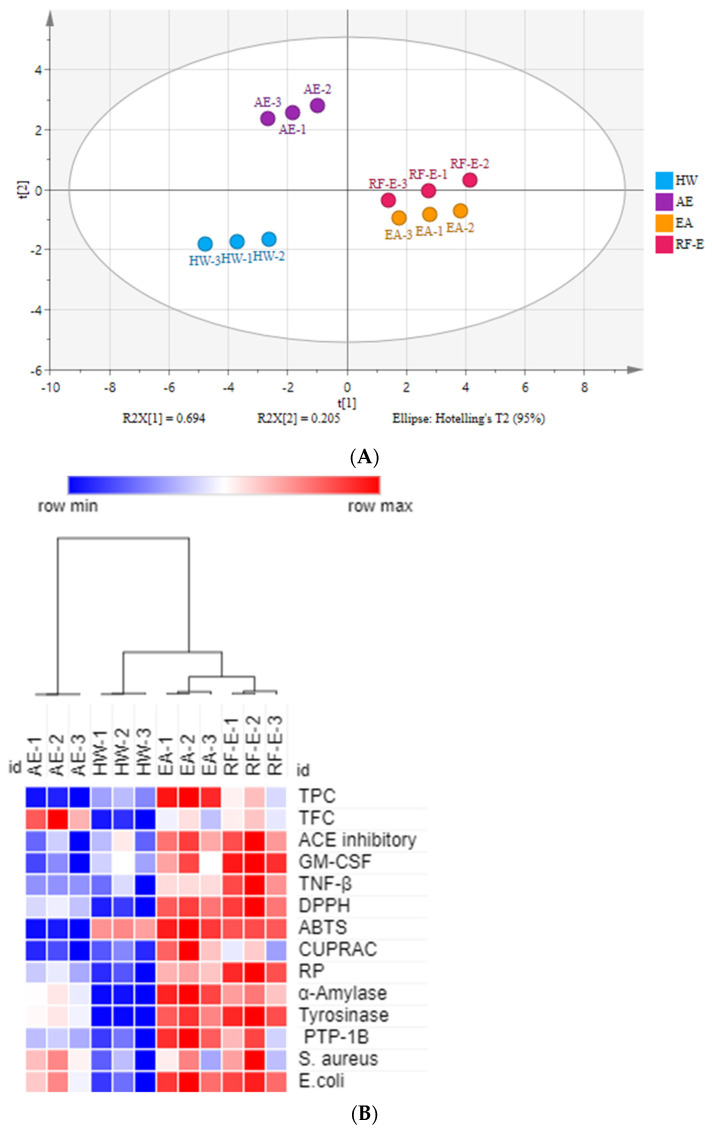
Multivariate analysis of *A. trifoliata* flower extracts. (**A**) PCA score plot, R^2^X = 99.6%, Q^2^ = 96.3%. (**B**) Heatmap analysis. (**C**) OPLS-DA plot, R^2^X = 96.2%, R^2^Y = 99.6%, Q^2^ = 99.1%. (**D**) Permutation plot of OPLS-DA, with intercepts of R^2^ and Q^2^ being 0.24 and –1.1, respectively. (**E**) S-plots associated with OPLS-DA scores of Class 1 and 2 members. See Figure 1 for abbreviations.

**Table 1 foods-11-03410-t001:** Non-anthocyanin polyphenols identified in *A. trifoliata* flowers extracts by UPLC-PDA-TOF-ESI-MS.

No.	RT	Mass Error	[M-H]^-^	Fragment Ions	Formula	Suggested Compounds	Response				Reference/Standard
	(min)	(ppm)					RF-E	EA	AE	HW	
1	3.79	−0.5	353	191, 135, 131	C_16_H_18_O_9_	Chlorogenic acid	126,206	128,733	71,463	83,772	Standard
2	4.16	0	353	201, 173, 135, 93	C_16_H_18_O_9_	Cryptochlorogenic acid	62,857	61,068	48,717	56,961	[26]
3	4.24	0.1	353	191, 135, 131	C_16_H_18_O_9_	Neochlorogenic acid	188,402	185,048	123,760	138,942	[26]
4	4.26	0.3	367	191, 135, 93	C_17_H_20_O_9_	Methyl 3-caffeoylquinate	14,443	11,797	6043	7380	[25]
5	4.47	−0.4	179	135	C_9_H_8_O_4_	Caffeic acid	9373	8474	5642	6592	Standard
6	5.06	0.5	367	191, 161	C_17_H_20_O_9_	3-feruloylquinic acid	7987	6995	3181	4042	Standard
7	5.28	−0.5	755	575, 413, 284, 161	C_33_H_40_O_20_	Quercetin 3-O-(2,6-α-L-dirhamno-pyranosyl)-β-D-galactopyranoside	7456	7794	2041	2653	[27]
8	5.56	0.8	163	119	C_9_H_8_O_3_	p-coumaric acid	1573	1220	878	899	[28]
9	5.67	−1.3	609	429, 284, 227	C_27_H_30_O_16_	Quercetin 3-O-robinobioside,	28,035	26,725	8604	6329	[29]
10	5.92	−1.4	609	477, 301, 271, 161	C_27_H_30_O_16_	Quercetin-3-rutinoside	39,759	38,015	13,955	16,940	[27]
11	6.26	−1.7	463	301, 271	C_21_H_20_O_12_	Quercetin-3-O-glucoside	12,350	12,659	1906	2903	[25]
12	6.94	−0.9	515	375, 353, 335, 191, 179, 161	C_25_H_24_O_12_	Isochlorogenic acid	183,785	176,185	48,219	55,802	Standard
13	7.02	−1	609	301, 227, 151	C_27_H_30_O_16_	Quercetin 3-O-neohesperidoside	80,762	75,630	15,931	22,194	[29]
14	7.24	−0.8	447	285, 255, 151	C_21_H_20_O_11_	Kaempferol-3-O-galactopyranoside	24,253	23,487	3459	4552	[27]
15	10.05	−0.7	301	/	C_15_H_10_O_7_	Quercetin	2629	4694	33,129	24,271	Standard
16	11.31	−0.3	285	227, 211, 185	C_15_H_10_O_6_	Kaempferol	46,545	58,540	75,225	53,514	Standard
						Hydroxicinnamic acids	594,626	579,520	307,903	354,390	
						Flavonols	241,789	247,544	154,250	133,356	
						Total non-anthocyanin polyphenols	836,415	827,064	462,153	487,746	

HW, hot water extraction; AE, acidified EtOH extraction; EA, enzymatic-assisted extraction; RF-E, radio frequency heating-assisted enzymatic extraction.

**Table 2 foods-11-03410-t002:** Correlation analysis between non-anthocyanin polyphenols with biological activities of *A. trifoliata* flower extracts.

Non-Anthocyanin Polyphenols ≠	ACE Inhibitory	GM-CSF	TNF-β	DPPH	ABTS	CUPRAC	RP	α-Amylase	Tyrosinase	*PTP-1B*	*S. aureus*	*E. coli*
1	**	*	-	-	-	*	-	-	-	-	-	-
2	*	**	-	-	*	-	-	-	-	-	-	-
3	***	**	*	-	-	-	-	-	-	-	-	-
4	**	**	**	-	-	-	-	-	-	-	-	-
5	***	**	*	-	-	-	-	-	-	-	-	-
6	***	**	*	-	-	-	-	-	-	-	-	-
7	**	*		-	-	*	-	-	-	-	-	-
8	*	**	***	-	-	-	*	-	-	-	-	-
9	**	-	*	*	-	-	*	-	*	*	-	-
10	***	*	*	-	-	-	-	-	-	-	-	-
11	**	*		-	-	*	-	-	-	-	-	-
12	**	*	*	*	-	-	-	-	-	-	-	-
13	**	*	*	-	-	-	-	-	-	-	-	-
14	**	*	*	*	-	-	-	-	-	-	-	-
15	***	**		-	-	-	-	-	-	-	-	-
16	-	-	-	-	-	-	-	-	-	-	-	-
Hydroxycinnamic acids	***	*	*	-	-	-	-	-	-	-	-	-
Flavonols	*	-	*	**	-	-	*	*	*	*	-	*
Total non-anthocyanin polyphenols	**	*	*	*	-	-	-	-	-	-	-	-
Total phenolics	-	-	-	-	-	*	-	-	-	-	-	-
Total flavonoids	-	-	-	-	-	-	-	-	-	-	**	-

*, *p* < 0.10; **, *p* < 0.05; ***, *p* < 0.01. ≠ Numbers 1–16 represent the non-anthocyanin polyphenols identified in Table 1. Data of DPPH (1, 1-diphenyl-2-picrylhydrazyl), ABTS (2, 2′-azino-bis-3-ethylbenzthiazoline-6-sulphonic acid), CUPRAC (cupric ion reducing activity), RP (Reducing power), α-Amylase, Tyrosinase, and PTP-1B (protein tyrosine phosphatase 1B) were referred to our previous study [1].

## Data Availability

Data is contained within the article.

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
