# Peer review of "Evaluation of Radio Frequency-Assisted Enzymatic Extraction of Non-Anthocyanin Polyphenols from Akebia trifoliata Flowers and Their Biological Activities Using UPLC-PDA-TOF-ESI-MS and Chemometrics"

_foods, 2022, doi:10.3390/foods11213410_

Round 1
Reviewer 1 Report
The manuscript is very interesting and in a great part absolutely original, only following a previous more general study of the same research group on the use of Akebia trifoliata flowers as a source of polyphenols, flavonoids and anti-microbial agent.
The authors compare four different extraction methods, using different solvents, concentration and contact times.
They study in deep the chemical, biochemical and anti-microbial characteristics of the extracts, using diverse determination techniques and comparing their results using different statistical tools.
Presentation and discussion of results is detailed and adequate.
I have only one quotation: As the authors know a great number of techniques for component extraction of fresh vegetables (flowers, leaves, fruits, roots) are done on the whole material (probably partially pre-treated - cut in pieces, deskinned, etc. -). The authors freeze-dry the fresh flowers and later extract the freeze-dried material. Besides being an expensive and little practical method for industrial/commercial level processing, freezing and later heating during the final desorption stage can provoke unaccounted changes in components.
I know perfectly that this way is more practical for preparing samples for further studies, I see every day to do so in our research institute. But results may not be the same.
In page 11: “As shown in Fig 1B, all our extracts showed a large absorption peak…”. Is it Fig 3B?
Author Response
I have only one quotation: As the authors know a great number of techniques for component extraction of fresh vegetables (flowers, leaves, fruits, roots) are done on the whole material (probably partially pre-treated - cut in pieces, deskinned, etc. -). The authors freeze-dry the fresh flowers and later extract the freeze-dried material. Besides being an expensive and little practical method for industrial/commercial level processing, freezing and later heating during the final desorption stage can provoke unaccounted changes in components. I know perfectly that this way is more practical for preparing samples for further studies, I see every day to do so in our research institute. But results may not be the same.
R: Thanks for this comment. In this study, our purpose is to study active ingredients such as polyphenols in flowers, and we hope to reduce the destruction of raw materials as much as possible. Compared to the conventional hot air drying, freeze-drying is the best method of water removal with final products of highest quality despite high costs and low in-field practicability. Thus, we choose to use freeze drying to minimize raw material damage. In future commercial applications, it is indeed necessary to replace freeze-drying with cheaper drying technologies and optimize the relevant operating parameters. We hope that our responses meet your expectations.
In page 11: “As shown in Fig 1B, all our extracts showed a large absorption peak…”. Is it Fig 3B?
R: Thank you so much for pointing out this mistake. The figure number has been changed to “Fig. 3B” as line 233.

Reviewer 2 Report
The manuscript has been reviewed now and the author has done Evaluation of Radio Frequency-assisted Enzymatic Extraction of Non-anthocyanin Polyphenols from Akebia trifoliata Flowers and Their Biological Activities Using UPLC-PDA-TOF-ESIMS and Chemometrics. The following points need to be addressed for the consideration of the manuscript
The abstract should contain key findings of research with conclusive lines
Material and methods should be descriptive, especially in sections 2.2, 2.3 and 2.6
The result and discussion part need to improve author has referred to their previous study in most of the discussion part. The mechanism for biological activities should be included in the antimicrobial anti-inflammatory section
You can follow latest publication for more detail
- DOI:
- 10.3390/gels8070434
Rewrite the conclusion
References are not written in the format of the journal

Author Response
The abstract should contain key findings of research with conclusive lines.
R: Thanks for this comment. The abstract has been revised as suggested in line 10-12 and line 17-19.
Material and methods should be descriptive, especially in sections 2.2, 2.3 and 2.6
R: Thanks for this comment. Considering the suggestion of reviewer, the sufficient details of methods have been added in the text in line 67-87, line 90-95, line 98-101, line 133-138, line 144-146.
The result and discussion part need to improve author has referred to their previous study in most of the discussion part. The mechanism for biological activities should be included in the antimicrobial anti-inflammatory section.
R: Thanks for this comment. Because this paper is a continuation of the previous research paper (Radio frequency-assisted enzymatic extraction of anthocyanins from Akebia trifoliata (Thunb.) Koidz. flowers: Process optimization, structure and bioactivity determination), there are many points in the Results and Discussion sections that need to be compared with the previous paper. Considering the suggestion of reviewer, we have carefully revised the Results and Discussion sections in line 188-189, line 194, line 198-199, line 236-238, line 254-257, line 254-257, line 327-329. The discussions about mechanism for biological activities were showed in the antimicrobial anti-inflammatory section such as line 259-260, line 276-283, line 276-283, 291-295, line 300-302, line 318-321
Rewrite the conclusion
R: Thanks for this comment. The conclusion has been revised as suggested in line 395-396 and line 398-400.
References are not written in the format of the journal.
R: Thanks for this comment. The formats of references have been revised in accordance with this journal.
Additionally, we have sought a help from native speaker to improve the English. And we have carefully read the manuscript and revised all possible spelling mistakes.

Round 2
Reviewer 2 Report
The author has incorporated the desired changes in the manuscript therefore, the article is now suitable for publication in the journal